# Psychological Profiles of Treatment-Seeking Adults with Overweight and Obesity: A Cluster Analysis Approach

**DOI:** 10.3390/jcm11071952

**Published:** 2022-03-31

**Authors:** Natalija Plasonja, Anna Brytek-Matera, Greg Décamps

**Affiliations:** 1Department of Human Sciences, Faculty of Psychology, Université de Bordeaux, LabPsy, EA 4139, F-33000 Bordeaux, France; greg.decamps@u-bordeaux.fr; 2Institute of Psychology, University of Wroclaw, 50-527 Wroclaw, Poland; anna.brytek-matera@uwr.edu.pl

**Keywords:** hierarchical cluster analysis, obesity, psychological profile, appearance, well-being, depression

## Abstract

Background: Overweight and obesity are associated with depression and well-being. Some psychological characteristics play a role in explaining well-being and depression in obesity and in identifying specific patient profiles. However, subtyping individuals with overweight/obesity based on variables like self-esteem or stress has not often been done. Therefore, our objective was to explore the psychological profiles of treatment-seeking individuals overweight or with obesity and to compare their depression and well-being. Methods: Data regarding eating self-efficacy, well-being, depression, physical hunger, self-esteem, body satisfaction and perceived stress in individuals with overweight/obesity were collected from the ESTEAM cohort. Hierarchical cluster analysis and mean comparisons were performed on female (*n* = 1427) and male samples (*n* = 310). Results: Three psychological profiles were identified in both samples. The “High psychological concerns” profile and the “Low psychological concerns” profile were identical in both samples. The third profile, “Bodily concerns”, differed by sex and was characterized by appearance dissatisfaction for women and by appearance and eating concerns for men. The ”Low psychological concerns” profile presented the highest well-being and the lowest depression scores in both samples. Discussion: The findings support the hypothesis of the heterogeneity of individuals with overweight and obesity and suggest sex-related therapeutic approaches.

## 1. Introduction

Overweight and obesity are a major public health concern worldwide, with obesity rates having nearly tripled since 1975 [1]. Worldwide, 37.4% of the adult population is overweight, while 23.7% of adults suffer from obesity [2]. The prevalence of overweight and obesity among adults is higher in high-income Western countries when compared to low-income countries, with the overweight rate in the United States exceeding 32.5% and the prevalence of obesity estimated at 37.7% [3].

The etiology of obesity is multifactorial, combining biological, social and psychological aspects. One of them is the imbalance between energy intake and energy expenditure, mainly due to an increase in energy-dense food combined with a decrease in physical activity [1]. Other factors play an important role in the onset of obesity as well, such as the microbiome [4], sleep duration [5,6] and genes [7,8], as well as the socioeconomic status of the individual [9,10], but also obesogenic environments (e.g., proximity to fast food restaurants or gyms) [11,12,13,14], air pollution [15] and the lifestyle of the individual [8,16]. Numerous psychological correlates have been identified as predictors of obesity [17,18]. Many of them are interrelated and influence each other [19,20]. Like its determinants, the consequences of obesity are multifactorial and diverse. The physical consequences of obesity include diabetes, cardiovascular diseases, high blood pressure and some forms of cancer [8,21,22]. Social consequences include rejection and discrimination, compounded by internalized weight bias [18,23,24,25]. Obesity can also have psychological consequences, including depression, binge eating, high levels of stress and body dissatisfaction, low self-esteem, low well-being and poor quality of life [23,24,25,26,27].

Like overweight and obesity, depression affects a large part of the population. It is estimated to affect 3.8% of the global population (all ages and sexes combined), with 5% of the adult population and 5.7% of adults over 60 suffering from depression [28]. Many studies have explored the links between obesity and depression, and several similarities between the two conditions have been identified. Both are related to higher mortality rates [29,30] and to the nonadherence to medication, exercise or weight management programs [23,25,31]. Furthermore, it has been shown that the presence of one can precipitate the onset of the other. Compared to individuals with a Body Mass Index (BMI) in the normal range, studies have shown that overweight and obesity increase the risk of depression [18,23,24,32,33,34,35], with some authors describing a dose–response effect of obesity severity on the odds of depression [23,25,27,35]. For instance, Pereira-Mianda et al. [24] reported that overweight individuals had a 7% higher probability of suffering from depression, while individuals with obesity presented a 32% probability of suffering from depression.

Numerous meta-analyses have shown that the links between overweight, obesity and psychological factors are moderated by sex, such that women living with overweight or obesity are more likely to suffer from depression than men [18,23,24,25,32,33,36] and have a more impaired psychological state, characterized by low self-esteem, higher levels of anxiety and body dissatisfaction [18,23,35]. Age is another sociodemographic factor that influences the comorbidity between obesity and depression. Although studies exploring its moderating effects on the aforementioned relationship are somewhat rare, it was pointed out in a meta-analysis by Preiss et al. [25] that a younger age range (18–39 years or 25–65 years, depending on the studies) increases the risk of comorbid obesity and depressive symptoms. Other correlates increasing the risk of the cooccurrence of obesity and depression concern physical health. Several studies have shown that both conditions are associated with biological and medical factors [24,25,37,38,39,40,41,42].

Even if certain risk factors for the development of depression in obesity have been identified, they do not explain the interindividual differences observed in people with overweight or obesity. Furthermore, these factors could have combined effects on the overall physical and mental health. Many authors have agreed on the heterogeneity of individuals with overweight or obesity [18,23,33,43] and have suggested that future studies should be devoted to identifying groups of individuals that display the highest risk of presenting these two conditions [23,44]. Therefore, the identification of different subgroups of individuals overweight or with obesity according to their biological, psychological and/or social characteristics could be useful for pursing the complexity of the correlates of obesity.

Cluster analysis is a technique that establishes patterns based on the shared characteristics of individuals, clustering them into mutually exclusive groups [45]. In this sense, cluster analysis can be helpful in identifying high-risk adults based on characteristics that can be addressed by psychological interventions. In addition to subtyping individuals according to their characteristics, cluster analysis could also provide guidance for later interventions for obesity prevention tailored to the needs of high-risk groups [18,25,46] and, as suggested by Markowitz et al. [23], target comorbid depression and obesity, rather than treat each condition separately. Cluster analysis is a widely used technique in the field of psychology [47,48,49,50]. However, in the field of obesity, as one of its causes is a misbalance between caloric intake and energy expenditure [26], the current focus in the literature is on the identification of patterns of physical activity or eating behavior in young individuals or adults living with overweight or obesity [51,52,53,54,55,56,57,58,59,60]. Up to now, there have been surprisingly few studies that have focused on identifying psychological profiles in adults with obesity using cluster analysis [61,62,63,64,65,66,67,68]. These studies have sought to subtype adults with overweight or obesity according to their dietary restraint [62], personality [61,63,65,68], affect [64], interpersonal functioning [66] or psychiatric comorbidities [67]. In some of these studies, high levels of depressive symptomatology have been observed in one or more identified profiles of individuals with overweight or obesity. However, it should be kept in mind that, in these studies, depressive symptomatology is based on self-reported measures and that not all individuals included in these profiles suffer from depression.

It is worth pointing out that, in the majority of these studies, the samples do not allow for comparisons between women and men, especially since some studies use exclusively female samples [61,62,64]. In the research that includes men, the male samples are quite small, which makes it impossible to conduct specific analyses on male and female subsamples. Furthermore, the majority of studies have focused on identifying profiles of individuals with overweight or obesity according to the stable characteristics of the participants (e.g., identity [69], personality [61], temperament [65,68] and personality disorders [63]), for which psychological interventions are not suitable. However, some studies have sought to identify psychological profiles on the basis of psychological correlates of obesity, such as negative affect and dietary restraint [62,64], while others have explored the quality of life of individuals with overweight and obesity [62,63,65,66]. Nonetheless, none of them explored their levels of well-being.

Consequently, the aim of the present study was to explore the psychological profiles of individuals living with overweight or obesity by taking into account multiple psychological correlates of obesity such as self-esteem, eating self-efficacy, perceived stress, physical hunger and body satisfaction and to compare their depressive symptomatology and well-being levels. As sex can influence the relationship between obesity and depression in such a way that women with overweight are more likely than men to develop depression [18,23,24,25,32,33,36], we decided to consider cluster solutions in separate female and male subsamples. The aim of this study was to explore the existence of psychological profiles specific to men and women with overweight or obesity, as well as to investigate differences in the age, BMI, depressive symptomatology and well-being levels between the profiles identified.

## 2. Methods

### 2.1. ESTEAM Cohort

The data used in this study were extracted from the ESTEAM cohort. The description of the cohort and additional information were provided in a recent study [70].

### 2.2. Participants

The total sample consisted of 1737 participants, 82% of whom were female. Participants were included if their age was between 18 and 64 years (mean age = 44.44, SD = 11.25 years) and if their BMI was equal or superior to 25 kg/m^2^, indicating a state of overweight (mean BMI = 32.29 ± 5.62; 25.0–67.5 kg/m^2^). Nine hundred and sixty-seven participants from the total sample (56%) responded to the depressive symptomatology scale, the Major Depression Inventory (MDI), in addition to other questionnaires used in the present study.

In the female subsample, 1427 women were included. The mean age of the women was 44.40 years (SD = 11.26; 18–64 years), and the mean BMI was 32.26 kg/m^2^ (SD = 5.63; 25.0–67.5 kg/m^2^). In this sample, 797 women (55.9% of the subsample) responded to the depressive symptoms scale.

The male subsample consisted of 310 men. Their mean age was 44.64 years (SD = 11.19; 18–64 years) and mean BMI was 32.47 kg/m^2^ (SD = 5.59; 25.26–55.66 kg/m^2^). Among the participants, 170 men (54.8% of the subsample) responded to the MDI.

### 2.3. Measures

#### 2.3.1. Eating Self-Efficacy

Eating self-efficacy was assessed using WEL-Fr-C [70], a French validation of the Weight Efficacy Life-Style Questionnaire (WEL) [71] intended for use in a clinical sample of individuals living with overweight or obesity. The psychometric properties of the scale are excellent. Its Cronbach’s alpha values, as well as those of all the scales used in the two subsamples, are displayed in Table 1. The eleven items of which the WEL-Fr-C is composed are assembled in two dimensions, evaluating eating self-efficacy in case of internal stimuli (items 1, 6, 11, 16 and 19 from the original scale) and eating self-efficacy in case of external stimuli (items 3, 7, 8, 12, 13 and 18 from the original scale). Items are scored on a Likert-type scale (0: Not confident, 9: Very confident), and a high score on the WEL-Fr-C is indicative of an important level of eating self-efficacy.

#### 2.3.2. State Self-Esteem

State self-esteem was assessed with the State Self-Esteem Scale (SSES) [72]. The original version of the SSES consists of 20 items, organized in three dimensions in order to evaluate performance self-esteem, social self-esteem and appearance self-esteem. A French version of the SSES was translated for use in the ESTEAM cohort. The exploratory, confirmatory and multigroup confirmatory factor analyses performed on the French SSES scale resulted in a 12-item version, organized in three dimensions. In this version, performance self-esteem was measured with items 1, 5, 9, 11, 14, 15 and 18 from the original scale*,* while social self-esteem was assessed with items 2, 4, 13 and 17 and appearance self-esteem with items 3, 7, 8 and 12 from the original scale. The items were scored on a Likert-type scale (1: Not at all, 5: Extremely). High-state self-esteem is characterized by a high score on the scale.

#### 2.3.3. Perceived Stress

Perceived stress was evaluated using the Perceived Stress Scale (PSS) [73], translated and validated in French [74]. The original version of this unidimensional scale consists of 14 items scored on a Likert-type scale (1: Never, 5: Very often). However, multigroup confirmatory factor analyses performed prior to this study yielded an 11-item version of the scale with satisfactory psychometric properties. The 11-item version was used in this study and was named PSS-11. A high score is indicative of a high level of perceived stress.

#### 2.3.4. Physical Hunger

Physical hunger was evaluated using a specific assessment tool created and validated by a group of medical experts to be used in the ESTEAM cohort [75]. The psychometric properties of the scale were satisfactory (The ten items of the scale explain 33% of the total variance. The goodness-of-fit indices were the following: *χ*^2^*/ddl* = 5.83; *RMSEA* = 0.068; *AGFI* = 0.938; *SRMR* = 0.039; *CFI* = 0.949; *TLI* = 0.929. The fidelity of the scale was satisfactory (*α* = 0.83; ICC = 0.87), and the sensibility was excellent (*δ* = 0.987; discrimination index > 0.40)) and its cross-sex and cross-BMI measurement invariances were confirmed using a multigroup confirmatory factor analysis (The multigroup confirmatory factor analysis indices were excellent (Δ*CFI* < 0.005, Δ*SRMR* < 0.010 and Δ*RMSEA* < 0.025) for configural, metric, scalar and strict invariance.). This unidimensional questionnaire consists of 10 Likert-type items (1: Never, 4: Often). A high score on the scale is indicative of a high level of physical hunger symptoms (e.g., “Uncontrollable hunger”, “Tremors of the extremities” and “Fatigue”).

#### 2.3.5. Body Satisfaction

Body satisfaction was assessed using the Body Image Questionnaire (BIQ) created and validated by Bruchon-Schweitzer [76,77]. The original version of this one-dimensional scale embodies 19 items, but after performing confirmatory factor analyses prior to this study, an 18-item version was retained. The items are scored on a five-point bipolar Visual Analogue Scale (1: Very much or often, 2: Somewhat or quite often, 3: In between or neither, 4: Somewhat or quite often and 5: Very much or often). A higher score indicates a higher level of body satisfaction.

#### 2.3.6. Well-Being

Well-being was evaluated with the World Health Organization’s Five Well-Being Index (WHO-5), created and validated in French by the Psychiatric Research Unit WHO Collaborating Centre in Mental Health [78]. The multigroup confirmatory analyses performed on the French version of the scale prior to this study confirmed its unidimensional structure and its excellent psychometric properties. The scale consists of five Likert-type items (0: At no time, 5: All of the time). High levels of well-being are characterized by a high score on the WHO-5 scale.

#### 2.3.7. Depressive Symptoms

Depressive symptoms were examined using a French adaptation of the Major Depression Inventory (MDI), created by the Psychiatric Research Unit [79] and translated specifically for the aims of this study, following standard scientific translation procedures [80]. Exploratory, confirmatory and multigroup confirmatory analyses performed on the French version of the scale confirmed its unidimensional 10-items structure and its satisfactory psychometric properties. The items are scored on a Likert-type scale (0: At no time, 5: All of the time). A higher score on the MDI scale is indicative of a greater severity of depressive symptomatology. This scale was administered only to participants who responded “At no time” or “Sometimes” on one of the five items of the WHO-5 scale or whose total WHO-5 scale score was inferior to 50.

#### 2.3.8. Anthropometric and Demographic Information

In addition to the aforementioned questionnaires, the physician reported the participant’s age, sex and his or her objectively measured weight (in kg) and height (in cm), which were used for the calculation of the BMI. World Health Organization criteria was used to determine overweight and obesity [81]. BMI values between 18.5 and 24.9 indicate normal weight, values between 25.0 and 29.9 indicate overweight, Grade I obesity is defined by BMI values between 30.0 and 34.9 and Grade II obesity is characterized by BMI values between 35 and 39.9, while grade III obesity is defined by a BMI of 40.0 or more.

### 2.4. Statistical Analyses

Identical statistical analyses were performed in both subsamples. The Grubbs test was used to investigate possible outliers. No outliers were detected in the present data. The normality of the data was then examined using the Kolmogorov–Smirnov test. Next, a two-step cluster analysis was performed on the z-scores of the variables included in the present study to identify the psychological profiles of women and men with overweight or obesity. The packages used for these analyses were {FactoMineR} and {factorextra}. Since the number of clusters was not known beforehand, an agglomerative hierarchical clustering based on the results of the principal components analysis was performed using Ward’s method [82]. The agglomerative technique considers each observation as a separate cluster. Clusters are then assembled according to their decreasing level of similarity [83]. The similarity method applied was the Euclidian distance. The optimal number of clusters to retain was determined based on the gain inertia, as well as the interpretability of the clusters. The predominant characteristics within each cluster were identified using the strength and direction of the average z-scores (−1;1). The k-means clustering analysis was performed in the second step to confirm the cluster solution. This analysis uses centroids to create distinct clusters. K-means clustering [84] classifies objects into multiple groups in such a way that cases in the same group are as similar as possible; that is, the intraclass similarity is maximized, while cases in different groups are as divergent as possible, meaning that the interclass similarity is minimized [85].

In order to investigate the internal validity of the clusters, nonparametric tests for mean comparisons were carried out using the Kruskal–Wallis test, followed by the Wilcoxon test for post-hoc analyses comparing the mean levels of the following clustering variables: External Stimuli and Internal Stimuli subscales from the WEL-Fr-C and Appearance, Social and Performance state self-esteem subscales, Physical hunger, BIQ and PSS-11. The external validity of the clusters was examined by comparing the mean levels of depression, well-being, BMI and age across different clusters. Statistical analyses were performed using RStudio software (RStudio Inc., Boston, MA, USA) for Windows, version 3.4.2.

## 3. Results

### 3.1. Descriptive Statistics of the Sample

The Cronbach’s alpha values of the scales used in the two subsamples are presented in Table 1.

Mean comparisons according to sex were performed on the total sample using the Wilcoxon test. No significant differences were observed between women and men on the variables used in this study. The descriptive statistics of the sample are displayed in Table 2.

### 3.2. Cluster Solution

#### 3.2.1. Female Subsample

The hierarchical cluster analysis yielded a three-cluster solution of women living with overweight or obesity based on their psychological characteristics.

The first cluster, in which 487 women were included (34% of the sample), was characterized by a significant level of psychological difficulties. This cluster had high Physical hunger and PSS-11 scores and low WEL-Fr-C, SSES and BIQ levels. The second cluster, composed of 629 women (44% of the sample), was defined by appearance dissatisfaction, since it displayed low SSES appearance scores and intermediate scores on the other scales. Finally, the third cluster, in which 311 women were included (22% of the sample), was described by the lack of significant psychological difficulties, as it displayed opposite tendencies to Cluster 1. Cluster 3 showed high levels of WEL-Fr-C, SSES and BIQ and low levels of Physical hunger and PSS-11. Considering the given characteristics of each cluster, they were named as follows: “High psychological concerns” profile (Cluster 1), “Bodily concerns” profile (Cluster 2) and ”Low psychological concerns” profile (Cluster 3). The cluster solution is presented in Figure 1.

#### 3.2.2. Male Subsample

The hierarchical cluster analysis also yielded a three-cluster solution of men living with overweight or obesity, according to their psychological characteristics (Figure 2). Cluster 1, composed of 83 men (27% of the sample) was, as in the female subsample, characterized by a significant level of psychological difficulties. This cluster had high Physical hunger and PSS-11 scores and low WEL-Fr-C, SSES and BIQ levels. Cluster 2, that included 152 men (49% of the sample), was characterized by eating and appearance concerns, as it presented low SSES appearance and eating self-efficacy scores. Cluster 3, in which 75 men were included (24% of the sample), was defined by a lack of significant psychological difficulties, since it displayed opposite tendencies to Cluster 1. Cluster 3 displayed high scores on the two dimensions of WEL-Fr-C, all the dimensions of SSES and BIQ and low scores for Physical hunger and PSS-11. Considering the characteristics of each cluster, the profiles identified in the male samples were given the same names as in the female samples: “High psychological concerns” profile (Cluster 1), “Bodily concerns” (Cluster 2) and “Low psychological concerns” profile (Cluster 3).

### 3.3. Internal Validation of the Clusters

#### 3.3.1. Female Subsample

Since the data were not normally distributed, a nonparametric Kruskal–Wallis test and a Wilcoxon post-hoc analysis were carried out to examine the differences between the clusters. The results showed that Cluster 1 had the lowest scores on the WEL-Fr-C subscales, SSES subscales and BIQ scale when compared to the other two (*p* < 0.001) and the highest scores for Physical hunger and PSS-11 (*p* < 0.001) (Table 3). Cluster 2 had higher scores on the WEL-Fr-C subscales, SSES subscales and BIQ scale compared to Cluster 1 (*p* < 0.001) but lower than Cluster 3 (*p* < 0.001). Additionally, Cluster 2 had lower scores on the Physical hunger and PSS-11 scales compared to Cluster 1 (*p* < 0.001) but higher than Cluster 3 (*p* < 0.001). Compared to the other two, Cluster 3 had the highest scores on the WEL-Fr-C subscales, SSES subscales and BIQ scale and the lowest Physical hunger and PSS-11 scores (*p* < 0.001). The size effect of these differences ranged from 0.15 to 0.49.

#### 3.3.2. Male Subsample

The nonparametric Kruskal–Wallis test and the Wilcoxon post-hoc analysis were applied for mean comparisons. The results showed that Cluster 1 had the lowest scores on the WEL-Fr-C subscales, SSES subscales and BIQ scale when compared to the other two (*p* < 0.001) and the highest Physical hunger and PSS-11 scores (*p* < 0.001) (Table 4). Cluster 2 had higher scores on the WEL-Fr-C subscales, SSES subscales and BIQ scale compared to Cluster 1 (*p* < 0.001) but lower than Cluster 3 (*p* < 0.001). Furthermore, Clusters 2 and 3 did not differ significantly on the SSES social subscale (*p* = 0.26). Cluster 2 had lower scores on the Physical hunger and PSS-11 scales compared to Cluster 1 (*p* < 0.001) but higher than Cluster 3 (*p* < 0.001). The size effect of these differences ranged from 0.34 to 0.55, except for the group differences on the Social state self-esteem score, for which the size effect was extremely small (*eta*^2^ < 0.01).

### 3.4. External Validation of the Clusters

#### 3.4.1. Female Subsample

The mean comparison analyses showed significant differences in the well-being and depressive symptomatology scales according to cluster (*p* < 0.01, *eta*^2^ = 0.40 and *eta*^2^
*=* 0.30, respectively; Table 5). Cluster 1 had the lowest well-being and the highest depressive symptomatology scores compared to the other two (*p* < 0.01). Cluster 2 showed higher well-being and lower depressive symptomatology scores when compared to Cluster 1 (*p* < 0.01) but lower well-being and higher depressive symptomatology scores than Cluster 3 (*p* < 0.01). Of the three clusters, Cluster 3 displayed the highest well-being score and the lowest depressive symptomatology score (*p* < 0.01). The different clusters did not differ in BMI values (*p* > 0.05). However, even though the Kruskal–Wallis test gave a significant difference in age between the clusters, its effect size was extremely small (*eta*^2^ = 0.006).

#### 3.4.2. Male Subsample

As in the female subsample, the mean comparisons revealed significant differences on the well-being and depressive symptomatology scales depending on the cluster (*p* < 0.01, *eta*^2^ = 0.44 and *eta*^2^ = 0.42, respectively; Table 6). Cluster 1 had the lowest well-being and the highest depressive symptomatology scores compared to the other two (*p* < 0.001). However, Cluster 2 displayed higher well-being and lower depressive symptomatology compared to Cluster 1 (*p* < 0.001) but lower well-being and higher depressive symptomatology compared to Cluster 3 (*p* < 0.001). Of the three profiles, Cluster 3 had the highest well-being and the lowest depressive symptomatology scores (*p* < 0.01). The different clusters did not differ in BMI values nor in age (*p* > 0.05).

## 4. Discussion

The main objective of this study was to identify the psychological profiles of individuals with overweight and obesity. In both female and male subsamples, a three-cluster solution was observed: “High psychological concerns” profile (Cluster 1), “Bodily concerns” (Cluster 2), characterized by appearance dissatisfaction in women and by appearance and eating concerns in men and “Low psychological concerns” profile (Cluster 3). In both subsamples, Clusters 1 and 3 were identical: Cluster 1 was defined by low levels of eating self-efficacy, self-esteem and body satisfaction and high levels of perceived stress and physical hunger, while Cluster 3 showed the opposite tendencies: high levels of eating self-efficacy, self-esteem and body satisfaction and low scores for perceived stress and physical hunger. Cluster 2 differed slightly between the two subsamples. In the female subsample, Cluster 2 was defined by low levels of appearance self-esteem and average scores on the other scales. However, in the male subsample, in addition to difficulties with appearance self-esteem, Cluster 2 also had low levels of eating self-efficacy. These findings are in line with those that argue that obesity outcomes differ by sex [18,23,35]. The cluster size was similar in the two subsamples. In both subsamples, Cluster 1 encompassed one-third of the sample (34% in the female subsample and 27% in the male subsample). Cluster 2 was the largest, including nearly half of the sample (44% and 49%, respectively). In both subsamples, Cluster 3 was the smallest, embodying less than a quarter of each subsample (22% and 24%, respectively).

In both subsamples, Cluster 1 had the highest levels of physical hunger and perceived stress and the lowest levels of eating self-efficacy, self-esteem and body satisfaction. The mean comparisons also revealed that Cluster 1 had the lowest well-being and the highest depression scores of the three profiles in both subsamples. One might have expected that the cluster with the lowest well-being and the highest level of depressive symptoms would have the highest BMI, in accordance with the literature. However, this is not the case in either sample, since the three clusters do not differ in their BMI.

Considering the low levels of depressive symptoms and the high well-being scores of the “Low psychological concerns” profile in both the female and male subsamples, a couple of hypotheses could be put forward to explain the observed results. One possible explanation could be the good coping skills of the individuals included in Cluster 3. When faced with a stressor, the adoption of good coping skills plays a fundamental role in the physical and psychological outcomes of the individual [86]. Coping skills are therefore protective of negative outcomes, such as stress and depression. Another possible explanation is that Cluster 3 may be mainly composed of individuals with metabolically healthy obesity (MHO). It is often characterized by the favorable metabolic profile, which means that this type of obesity is not accompanied by common metabolic disorders such as type 2 diabetes, high blood pressure or insulin sensitivity. Although there is no standardized definition of MHO, several characteristics are common to all MHO individuals, including low visceral mass, high insulin sensitivity and normal arterial blood pressure [87]. Individuals with MHO have lower levels of depressive symptoms [88], stress and anxiety [89,90] and higher well-being scores [90] than individuals with metabolically unhealthy obesity. Some studies have even found that individuals with MHO did not differ from the individuals with normal metabolically healthy weights in terms of the risk of depressive symptoms [88,89,90]. In their 16-year follow-up study, Hinnouho et al. concluded that the metabolic health status predicts depressive symptoms at the start of the follow-up but that obesity predicts a poorer evolution of depressive symptoms over time only in metabolically unhealthy individuals [88]. In both subsamples, Cluster 2 had intermediate well-being and depression scores. However, over time, this cluster may be at risk of developing the same psychological problems as the “High psychological concerns” profile. It would therefore be necessary to follow the three clusters over time to explore their evolution.

## 5. Clinical Implications

The results of our study corroborate the statements of other researchers who have confirmed the heterogeneity of individuals with obesity and have provided perspectives for future interventions [18,23,33,44]. Given the cluster solution identified in the two subsamples, it would be interesting to propose interventions tailored to the characteristics of each cluster. Similar characteristics to those observed in the “High psychological concerns” profile are often reported in the literature, and interventions aimed at reducing depression and overweight or obesity (e.g., cognitive behavioral therapy (CBT), enhanced CBT, enhanced focused CBT, behavioral weight loss treatment, acceptance and commitment therapy [91]) are intended for this profile of psychological functioning. While the management of weight loss and depression in obesity is beneficial, it may not be sufficient.

Individuals in the “Bodily concerns” profile may not be seen as high-risk patients. Nevertheless, this profile may be at risk of worsening their psychological conditions. It is therefore important to target their specific needs and adapt interventions according to the characteristics of the individuals included in this profile. Such interventions could prevent future negative psychological outcomes in Cluster 2, such as those observed in Cluster 1 (decreased well-being and increased depressive symptomatology). For instance, interventions incorporating body compassion were found to improve body concerns, restrictive eating, quality of life, life satisfaction and self-esteem [92,93,94]. They can therefore be recommended for individuals presenting appearance and/or eating concerns, as those included in Cluster 2. For men included in this profile, interventions targeting eating self-efficacy may also be beneficial. The “Low psychological concerns” profile may benefit from interventions that focus on maintaining a good quality of life and high well-being levels to prevent the negative evolution of this profile. Studies have shown that the most effective interventions for obesity are those that maintain a lifestyle intervention over an extended period of time [95], which is also pointed out in Obesity Canada’s guidelines for obesity management [96].

Finally, considering the high comorbidity between obesity and depression, the improvement of interventions in obesity is crucial. To do so, interventions need to target the factors responsible for the onset of depression in obesity, such as binge eating disorder, stigma, self-efficacy, and body dissatisfaction. Adding these elements to an intervention designed for the treatment of depression and obesity may enhance its effectiveness. In addition, this intervention can also be adapted for Cluster 2. Furthermore, it could be valuable to target the promotion of well-being and quality of life in obesity treatment, as shown in the Kg-free intervention by Palmeira et al. [97]. Testing the efficacy of these integrated interventions in comparison to the classical obesity treatment in a randomized controlled study could be useful in identifying the elements that contribute to improving the overall quality of life in individuals with overweight and obesity.

## 6. Limitations

There are limitations in the present study. Although we explored female and male cluster solutions separately, the female subsample was larger than the male subsample. This underrepresentation of men is common in studies on obesity [98,99]. In addition, given that data are provided by consultations with health professionals [70], the characteristics of the sample allow us to further identify the characteristics of individuals who seek treatment and who are, predominantly, women. The fact that our sample consisted of individuals seeking treatment is another limitation of our study. Some studies have found significant differences between individuals with overweight and obesity who seek treatment and those who do not seek treatment [23,25]. For example, individuals with obesity who sought nonsurgical treatment had a higher prevalence of depression, poorer health outcomes and more binge eating than those who did not seek treatment [23,25]. Therefore, our results cannot be generalized to individuals living with overweight or obesity who do not seek treatment. Another point to keep in mind is that the categorization of clusters may be explained by the presence of other unmeasured variables in the ESTEAM cohort, including some biological factors, like those used to define MHO or hypothalamic–pituitary–adrenal axis dysregulation, binge eating disorder, stigma, dieting and weight cycling, as presented in the comprehensive model of depression in obesity by Markowitz et al. [23].

## 7. Conclusions

In both women and men with overweight and obesity, three distinct profiles were identified on the basis of their psychological characteristics: the “High psychological concerns” profile, the “Bodily concerns” profile, characterized by appearance dissatisfaction in women and appearance and eating concerns in men and the “Low psychological concerns” profile. In both subsamples, the profiles differed significantly in depressive symptoms and well-being scores. We provided intervention perspectives and recommendations for each cluster. Further studies are needed to explore the existence of psychological profiles of individuals with overweight and obesity in samples including more men, as well as individuals not seeking treatment. Future studies should also focus on analyzing the evolution of each cluster over time.

## Figures and Tables

**Figure 1 jcm-11-01952-f001:**
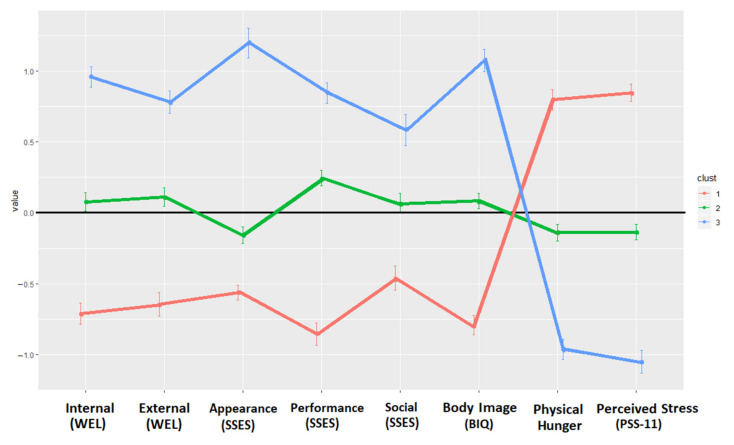
Graphical presentation of the clusters identified in the female subsample. Note. Internal (WEL): Resistance to internal stimuli; External (WEL): Resistance to external stimuli; Appearance (SSES): Appearance state self-esteem; Performance (SSES): Performance state self-esteem; Social (SSES): Social state self-esteem; Body Image (BIQ): Body Image Questionnaire; Perceived Stress (PSS-11): Perceived Stress Scale.

**Figure 2 jcm-11-01952-f002:**
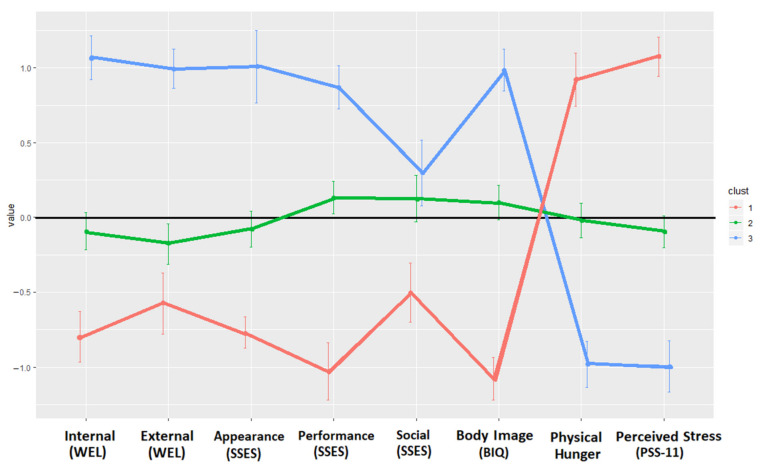
Graphical presentation of the clusters identified in the male subsamples. Note. Internal (WEL): Resistance to internal stimuli; External (WEL): Resistance to external stimuli; Appearance (SSES): Appearance state self-esteem; Performance (SSES): Performance state self-esteem; Social (SSES): Social state self-esteem; Body Image (BIQ): Body Image Questionnaire; Perceived Stress (PSS-11): Perceived Stress Scale.

**Table 1 jcm-11-01952-t001:** Cronbach’s alpha values of the scales used in the female and male subsamples.

Assessment Scale	Cronbach’s Alpha Values	
	Female Subsample (*n* = 1427)	Male Subsample (*n* = 310)
WEL-Fr-C	0.91	0.91
External i	0.89	0.88
Internal	0.91	0.91
SSES	0.84	0.84
Appearance	0.82	0.84
Social	0.79	0.79
Performance	0.82	0.84
MDI	0.83 *^a^*	0.81 *^b^*
WHO-5	0.88	0.89
Physical hunger	0.84	0.82
PSS-11	0.87	0.89
BIQ	0.89	0.91

Note. *^a^* sample of women that responded to the MDI scale (*n* = 797); *^b^* sample of men that responded to the MDI scale (*n* = 170). WEL-Fr-C: Weight Efficacy Life-Style Questionnaire; External: Resistance to external stimuli (WEL-Fr-C); Internal: Resistance to internal stimuli (WEL-Fr-C); SSES: State Self-Esteem Scale; MDI: Major Depression Inventory; WHO-5: World Health Organization-Five Well-Being Index; PSS-11: Perceived Stress Scale; BIQ: Body Image Questionnaire.

**Table 2 jcm-11-01952-t002:** Descriptive statistics and mean comparisons of the samples.

**Variable**	**Female Subsample** **(*n* = 1427)**	**Male Subsample** **(*n* = 310)**	***U* (1735)**
Age	44.4 (11.26)	44.64 (11.19)	217,560
BMI	32.26 (5.63)	32.47 (5.59)	215,376
WEL-Fr-C	58.49 (20.64)	59.33 (20.15)	217,556
External	34.22 (11.62)	34.22 (11.62)	219,442
Internal	24.27 (12.04)	24.84 (11.89)	215,583
SSES	43.94 (9.53)	44.95 (9.49)	206,266
Appearance	7.95 (3.92)	8.35 (3.95)	206,888
Social	25.08 (5.28)	25.39 (5.36)	213,725
Performance	10.91 (3.79)	11.21 (3.70)	209,931
WHO-5	51.24 (23.06)	52.39 (22.55)	215,230
Physical hunger	21.72 (6.29)	21.42 (5.89)	226,789
PSS-11	31.65 (7.87)	31.04 (7.82)	230,984
BIQ	55.85 (12.51)	56.85 (13.34)	209,886
**Variable**	**Female Subsample (*n* = 797)**	**Male Subsample** **(*n* = 170)**	***U* (965)**
MDI	23.56 (8.85)	22.69 (8.36)	71,550

Note. BMI: Body Mass Index; WEL-Fr-C: Weight Efficacy Life-Style Questionnaire; External: Resistance to external stimuli (WEL-Fr-C); Internal: Resistance to internal stimuli (WEL-Fr-C); SSES: State Self-Esteem Scale; Appearance: Appearance state self-esteem; Social: Social state self-esteem; Performance: Performance state self-esteem; WHO-5: World Health Organization-Five Well-Being Index; PSS-11: Perceived Stress Scale; BIQ: Body Image Questionnaire; MDI: Major Depression Inventory. Numbers in parentheses are standard deviations.

**Table 3 jcm-11-01952-t003:** Mean comparisons on the standardized clustering variables used in the female subsample.

**Assessment Scale**	**Cluster 1**	**Cluster 2**	**Cluster 3**	** *Kruskal-Wallis* **	** *Eta* ^2^ **
WEL-Internal	−0.71 (0.82) ^*a*^	0.08 (0.83) ^*b*^	0.96 (0.65) ^*c*^	535.49 ***	0.38
WEL-External	−0.64(0.94) ^*a*^	0.11 (0.84) ^*b*^	0.78 (0.70) ^*c*^	406.90 ***	0.28
SSES-Appearance	−0.56 (0.60) ^*a*^	−0.16 (0.75) ^*b*^	1.20 (0.95) ^*c*^	539.39 ***	0.43
SSES-Social	−0.46 (0.94) ^*a*^	0.07 (0.88) ^*b*^	0.58 (0.98) ^*c*^	201.43 ***	0.15
SSES-Performance	−0.86 (0.87) ^*a*^	0.24 (0.70) ^*b*^	0.85 (0.65) ^*c*^	629.49 ***	0.43
Physical hunger	0.80 (0.81) ^*a*^	−0.14 (0.76) ^*b*^	−0.96 (0.64) ^*c*^	624.01 ***	0.43
BIQ	−0.79 (0.75) ^*a*^	0.08 (0.72) ^*b*^	1.07 (0.71) ^*c*^	679.24 ***	0.47
PSS-11	0.85 (0.70) ^*a*^	−0.14 (0.71) ^*b*^	−1.05 (0.73) ^*c*^	721.57 ***	0.49

Note. WEL-Internal: Resistance to internal stimuli; WEL-External: Resistance to external stimuli; SSES-Appearance: Appearance state self-esteem; SSES-Social: Social state self-esteem; SSES-Performance: Performance state self-esteem; BIQ: Body Image Questionnaire; PSS-11: Perceived Stress Scale. *^a^*^, *b*, *c*^ Different letters in the same row indicate statistically significant differences. Numbers in parentheses are standard deviations. *** *p* < 0.001.

**Table 4 jcm-11-01952-t004:** Mean comparisons on the standardized clustering variables used in the male subsample.

	Cluster 1	Cluster 2	Cluster 3	*Kruskal-Wallis*	*Eta* ^2^
WEL-Internal	−0.80 (0.77) ^*a*^	−0.09 (0.77) ^*b*^	1.07(0.64) ^*c*^	137.73 ***	0.45
WEL-External	−0.58 (0.93) ^*a*^	−0.18 (0.85) ^*b*^	0.99 (0.56) ^*c*^	114.09 ***	0.34
SSES-Appearance	−0.77 (0.47) ^*a*^	−0.08 (0.74) ^*b*^	1.01 (1.05) ^*c*^	120.83 ***	0.41
SSES-Social	−0.50 (0.90) ^*a*^	0.13 (0.98) ^*b*^	0.30 (0.95) b, ^*c*^	31.34 ***	0.098
SSES-Performance	−1.03 (0.87) ^*a*^	0.13 (0.68) ^*b*^	0.87 (0.63) ^*c*^	145.87 ***	0.48
Physical hunger	0.92 (0.82) ^*a*^	−0.02 (0.72) ^*b*^	−0.98 (0.67) ^*c*^	143.94 ***	0.46
BIQ	−1.08 (0.65) ^*a*^	0.10 (0.71) ^*b*^	0.99 (0.61) ^*c*^	174.24 ***	0.55
PSS-11	1.08 (0.59) ^*a*^	−0.10 (0.67) ^*b*^	−0.99 (0.74) ^*c*^	174.24 ***	0.55

Note. WEL-Internal: Resistance to internal stimuli; WEL-External: Resistance to external stimuli; SSES-Appearance: Appearance state self-esteem; SSES-Social: Social state self-esteem; SSES-Performance: Performance state self-esteem; BIQ: Body Image Questionnaire; PSS-11: Perceived Stress Scale. *^a^*^, *b*, *c*^ Different letters in the same row indicate statistically significant differences. Numbers in parentheses are standard deviations. *** *p* < 0.001.

**Table 5 jcm-11-01952-t005:** Mean comparisons on the well-being, depression and sociodemographic data in the female subsample.

Variable	Cluster 1	Cluster 2	Cluster 3	*Kruskal-Wallis*	*Eta* ^2^
Age	44.13 (11.09) ^a^	43.81 (11.66) ^*a,b*^	46.02 (10.58) ^c^	7.67 *	0.006
BMI	32.26 (5.59)	32.44 (5.77)	31.86 (5.39)	1.79	
WHO-5	33.82 (18.46) ^a^	53.83 (18.20) ^b^	73.30 (16.16) ^c^	574.56 **	0.40
MDI	27.96 (8.01) ^a^	19.68 (6.76) ^b^	13.53 (6.75) ^c^	242.41 **	0.30

Note. BMI = Body Mass Index; WHO-5: World Health Organization-Five Well-Being Index; MDI: Major Depression Inventory. *^a^*^, *b*, *c*^ Different letters in the same row indicate statistically significant differences. Numbers in parentheses are standard deviations. * *p* < 0.05 and ** *p* < 0.01.

**Table 6 jcm-11-01952-t006:** Mean comparisons on the well-being, depression and sociodemographic data in the male subsample.

	Cluster 1	Cluster 2	Cluster 3	*Kruskal-Wallis*	*Eta^2^*
Age	43.16 (10.75)	45.02 (11.10)	45.49 (11.84)	2.79	
BMI	33.66 (6.76)	31.99 (4.91)	32.12 (5.32)	2.86	
WHO-5	31.05 (16.23) *^a^*	54.08 (17.62) *^b^*	72.53 (16.46) *^c^*	132.86 ***	0.44
MDI	28.69 (6.59) *^a^*	18.43 (6.38) *^b^*	15.21 (5.52) *^b^*	71.69 ***	0.42

Note. BMI = Body Mass Index; WHO-5: World Health Organization-Five Well-Being Index; MDI: Major Depression Inventory. *^a^*^, *b*, *c*^ Different letters in the same row indicate statistically significant differences. Numbers in parentheses are standard deviations. *** *p* < 0.001.

## Data Availability

The data analyzed in this study cannot be shared publicly due to privacy concerns. The data access requests should be addressed to R&D Innovation Director at PiLeJe, Michel Dubourdeaux (m.dubourdeaux@pileje.com).

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
