# Peer review of "Psychological Profiles of Treatment-Seeking Adults with Overweight and Obesity: A Cluster Analysis Approach"

_jcm, 2022, doi:10.3390/jcm11071952_

Round 1

Reviewer 1 Report

This study revealed a psychological profiles of treatment-seeking adults with overweight and obesity and as the statistical method they used a cluster analysis approach. Overall, the merit of this study is the high sample size and the limitation of this study is not using Latent Profile/Class Analysis. I suggest the authors instead cluster analysis method they use latent profile/class analysis which is more robust method. please find my detailed comments below:

Abstract

page 1, lines 25-26: Remove this and discuss/conclude your results

Introduction

page 1, line 32-33: If there is more recent reference please replace it with your reference number 1.
Page 1,line 33: it is better to write higher rather than "is very high"

page 3, line 113: please change the word "few" to "a few"
page 3, lines 101-148: this part is more appropriate for discussion. Please amend this part and note that you are writing your introduction rather than discussion. 
page 3, lines 149-151: please note that your sample has the same limitation. So, please amend this part. 

page 4, lines 152-157: please add references. 
page 4, lines 162-165: please add references. And I suggest not to write this. Please simply do your statistics separately for men and women and there is no need to write study 1 and 2. 

Methods
page 4, lines 174-185: please describe how did you measure weight and height, and according to which reference you defined your overweight and obesity? Please add reference for MDI and reliability and validity.
Please add validity and reliability for the questionnaires you used in your study.

Page 6, line 248: please change Shapiro test to Kolmogrove Smironove. For high samples it is better to use Kolmogrove Smironove.

page 6, statistics: please mention which packages did you use?

Please add a descriptive table for your participants and compare their results. 

Author Response

This study revealed a psychological profiles of treatment-seeking adults with overweight and obesity and as the statistical method they used a cluster analysis approach. Overall, the merit of this study is the high sample size and the limitation of this study is not using Latent Profile/Class Analysis. I suggest the authors instead cluster analysis method they use latent profile/class analysis which is more robust method. please find my detailed comments below:

Authors’ response: Thank you for your kind feedback! We did not use LCA/LPA analyses because we have no prior theory allowing us to predict cluster solutions and because we are not specifically expecting to observe a latent variable.

Abstract

page 1, lines 25-26: Remove this and discuss/conclude your results

 Authors’ response:  As suggested, we have modified the abstract and have taken into account your remarks.

Introduction

page 1, line 32-33: If there is more recent reference please replace it with your reference number 1.

Authors’ response: Unfortunately, the WHO page on Obesity has not been updated since June 2021 and there are not any recent epidemiological studies on the prevalence of Obesity.

Page 1, line 33: it is better to write higher rather than "is very high"

Authors’ response: Following your suggestion, clarifications were provided in this sentence and your proposition has been included (lines 35-37).

page 3, line 113: please change the word "few" to "a few"

Authors’ response: Thank you for your proposition. Unfortunately, it was not approved by the translator who proofread the final version of the article.

page 3, lines 101-148: this part is more appropriate for discussion. Please amend this part and note that you are writing your introduction rather than discussion.

Authors’ response: Following yours and the second reviewer’s suggestions, we have simplified the Introduction and, hence, this paragraph as well.

page 3, lines 149-151: please note that your sample has the same limitation. So, please amend this part.

Authors’ response: Thank you for pointing this out! We have modified this part accordingly (lines 110-121).

page 4, lines 152-157: please add references.

Authors’ response: This paragraph has been modified and the references of the studies exploring different profiles have been added (lines 113-121).

page 4, lines 162-165: please add references.

Authors’ response: References regarding the higher risk of developing depression for women have been added (lines 127-129).

And I suggest not to write this. Please simply do your statistics separately for men and women and there is no need to write study 1 and 2.

Authors’ response: Thank you for your advice! The parts and mentions regarding Study 1 and Study 2 have been removed from the manuscript. The Methods, Results and Discussion sections have been corrected accordingly.

Methods

page 4, lines 174-185: please describe how did you measure weight and height, and according to which reference you defined your overweight and obesity?

Authors’ response: Thank you for suggesting these changes. The procedure for collecting weight and height data has been further described in paragraph “2.3.8. Anthropometric and demographic information” (lines 220-228).

Please add reference for MDI and reliability and validity.

Authors’ response: The MDI was translated to French by a group of doctors and for the use in the ESTEAM cohort. It has been validated by the authors of this article for the means of this study and for the use in the cohort, but the French version of the MDI has not been published.

Please add validity and reliability for the questionnaires you used in your study.

Authors’ response: We were only able to explore the internal consistency of the scales using the Cronbach’s alpha because the design of this study was cross-sectional. The internal consistency has been presented in Table 1. However, the different scales used in our study do not allow us to test their validity, other than by exploratory, confirmatory and multigroup confirmatory analyses. These information have been added in the “2.3. Measures” section.

Page 6, line 248: please change Shapiro test to Kolmogrove Smironove. For high samples it is better to use Kolmogrove Smironove.

Authors’ response: Thank you for your suggestion! We have run the normality analysis using the K-S test and have changed its name in the “2.4. Statistical analyses” section (line 232). The results were the same, i.e. data were not distributed normally.

page 6, statistics: please mention which packages did you use?

Authors’ response: The names of the packages were added in the section “2.4 Statistical analyses” (lines 235-236).

Please add a descriptive table for your participants and compare their results. 

Authors’ response: The descriptive statistics of the total sample were presented in a new table, Table 2, and their mean comparisons were addressed in “3.1 Descriptive statistics of the sample”, a segment added to the Results section. (lines 267-274).

Reviewer 2 Report

The authors explore psychological profiles in obese women and men, taking into account psychological correlates i.e. self-esteem and body satisfaction. Further the clusters have been compared in terms of depressive symptomatology and well-being. In general, this is an interesting approach to an important topic. It also has potential to provide clues for an individualised treatment approach to obesity. 

Still, I have a few comments and suggestions:  

Abstract: The detailed differences of well-being and depression between the clusters are not stated (line23-24).  Please insert. Perhaps also mention the names of the clusters again in the "discussion" for better understanding. Also, I find the discussion in the abstract a bit too general, maybe add 1-2 sentences.

Introduction: It seems a little bit too long and would like to see more stringency in it. Perhaps it should also be elaborated more what is new and different in this study. 

Method: I find the methods section confusing. I would suggest listing both studies under "Method" and the corresponding subheadings so that the reader can immediately compare them when reading.

Line 213-220: Here I would like to see some notes to the validity of the new test instrument. 

Figure 1 and 2 are not easy to read. Please use larger axis labels and thicker lines.

Line 402-406: Under item 3.3 "Discussion", this seems to be a summary of the results rather than part of the discussion. Untitle this subtitle or insert in appropriate place. 

Author Response

The authors explore psychological profiles in obese women and men, taking into account psychological correlates i.e. self-esteem and body satisfaction. Further the clusters have been compared in terms of depressive symptomatology and well-being. In general, this is an interesting approach to an important topic. It also has potential to provide clues for an individualised treatment approach to obesity.

Authors’ response: Thank you!

Still, I have a few comments and suggestions: 

Abstract:

The detailed differences of well-being and depression between the clusters are not stated (line23-24).  Please insert. Perhaps also mention the names of the clusters again in the "discussion" for better understanding. Also, I find the discussion in the abstract a bit too general, maybe add 1-2 sentences.

Authors’ response: Following yours and first reviewer’s suggestionq, the abstract has been changed and your remarks have been taken into account.

Introduction:

It seems a little bit too long and would like to see more stringency in it. Perhaps it should also be elaborated more what is new and different in this study.

Authors’ response: Thank you for your proposition. The introduction has been simplified, and the information on what our study adds to the literature have been provided (lines 110-133).

Method:

I find the methods section confusing. I would suggest listing both studies under "Method" and the corresponding subheadings so that the reader can immediately compare them when reading.

Authors’ response: Thank you for your advice. The parts and mentions regarding Study 1 and Study 2 have been removed from the manuscript. The Methods, Results and Discussion sections have been corrected accordingly.

Line 213-220: Here I would like to see some notes to the validity of the new test instrument.

Authors’ response: We are going to give a poster presentation regarding the Physical hunger scale validation at the end of March. We added the reference to the poster as well as footnotes about the information regarding the results from the EFA, CFA and MG-CFA conducted on the scale.

Figure 1 and 2 are not easy to read. Please use larger axis labels and thicker lines.

Authors’ response: Thank you for your suggestion! We have taken yours and the third reviewer’s comments into account and have modified the figures accordingly.

Line 402-406: Under item 3.3 "Discussion", this seems to be a summary of the results rather than part of the discussion. Untitle this subtitle or insert in appropriate place.

Authors’ response: Thank you for your recommendation. Since the results from both subsamples have been presented together, this part of the manuscript has been deleted.

Reviewer 3 Report

Thank you for the opportunity to review this manuscript, which uses cluster analysis to identify psychological profiles of men and women with overweight and obesity, and to compare these clusters on depressive symptoms and well-being. This manuscript was clearly written and fills a gap in the literature. It also has important clinical implications, including for treatment tailoring, which the authors describe. My feedback for ways to improve the manuscript is outlined below and mostly focuses on ensuring that the manuscript does not reinforce obesity or mental health stigma.

Overall, the authors do a nice job of avoiding stigmatizing language or a stigmatizing tone. However, given the ubiquity of obesity and mental health stigma and, thus, the need to be very cautious in how research frames obesity and mental health, I have included a few suggestions below to further decrease the possibility of stigmatization.

First, I would consider changing the names of the "unfavourable" and "favourable" clusters to something more descriptive and objective. I worry that "unfavourable" has negative connotations that might reflect on the individuals in this cluster, rather than on the difficulties that they are experiencing. Likewise, favourable has a tone of superiority that might position individuals in that cluster as "better" than individuals in the unfavourable cluster.

The authors do a nice job of making clear that obesity has multifaceted causes, which helps to reduce stigma around obesity. However, I would recommend rewording the beginning of this sentence (lines 37-38), "Even though the main cause of obesity is the imbalance between energy intake and energy expenditure, mainly due to an increase in energy-dense food combined with a decrease in physical activity [1]." As currently written, it may reinforce obesity stigma (i.e., the idea that those with obesity are to "blame" for their weight status and "just" need to change their behaviors to reduce their weight." I would instead include the imbalance between energy intake and energy expenditure as one of the multifactorial causes of obesity without highlighting it as the "main" cause. 

Line 57: Please clarify what you mean by "non-adherance to treatment." What treatment? Obesity treatment? Depression treatment? Medical care more generally? Also, to reduce stigma, I recommend something like "difficulty adhering to treatment recommendations" in order to reduce blame towards the person. 

Line 58: The authors generally do a really nice job of using person-first language in this manuscript. Although "eutrophic" is not a stigmatizing term in the same way that "obese" can be, I would still use person-first language here, e.g., "Compared to individuals with a BMI in the normal range."

In the paragraph starting on Line 101 that describes prior cluster analyses, I would be cautious about making it sound like all individuals with overweight or obesity have significant psychopathology. I might add some nuance to this paragraph to clarify that this is not the case and to reiterate that the reason for examining these clusters is to ensure that individuals with obesity with various psychological needs are adequately supported. 

Other comments:

In Figures 1 and 2, it would be helpful if the labels matched the way that you described the measures in the text (e.g., Calling "App" SSES-app or something like that). Otherwise, it's a bit hard to switch between the figure and the text.

Author Response

Thank you for the opportunity to review this manuscript, which uses cluster analysis to identify psychological profiles of men and women with overweight and obesity, and to compare these clusters on depressive symptoms and well-being. This manuscript was clearly written and fills a gap in the literature. It also has important clinical implications, including for treatment tailoring, which the authors describe. My feedback for ways to improve the manuscript is outlined below and mostly focuses on ensuring that the manuscript does not reinforce obesity or mental health stigma.

Overall, the authors do a nice job of avoiding stigmatizing language or a stigmatizing tone. However, given the ubiquity of obesity and mental health stigma and, thus, the need to be very cautious in how research frames obesity and mental health, I have included a few suggestions below to further decrease the possibility of stigmatization.

Authors’ response: Thank you for your feedback!

First, I would consider changing the names of the "unfavourable" and "favourable" clusters to something more descriptive and objective. I worry that "unfavourable" has negative connotations that might reflect on the individuals in this cluster, rather than on the difficulties that they are experiencing. Likewise, favourable has a tone of superiority that might position individuals in that cluster as "better" than individuals in the unfavourable cluster.

Authors’ response: Thank you for pointing this out and for your suggestions. We have changed the name of the profiles as follows: “Unfavourable functioning profile” to “High psychological concerns”, “Appearance dissatisfaction”/ “appearance concerns” to “Bodily concerns” and “Favourable functioning profile” to “Low psychological concerns”.

The authors do a nice job of making clear that obesity has multifaceted causes, which helps to reduce stigma around obesity. However, I would recommend rewording the beginning of this sentence (lines 37-38), "Even though the main cause of obesity is the imbalance between energy intake and energy expenditure, mainly due to an increase in energy-dense food combined with a decrease in physical activity [1]." As currently written, it may reinforce obesity stigma (i.e., the idea that those with obesity are to "blame" for their weight status and "just" need to change their behaviors to reduce their weight." I would instead include the imbalance between energy intake and energy expenditure as one of the multifactorial causes of obesity without highlighting it as the "main" cause.

Authors’ response: Thank you for mentioning this. We have modified this paragraph (lines 39-42) as well as the sentence on lines 97 and 98, by stating that the misbalance between caloric intake and energy expenditure is one of the causes of obesity, rather than its main cause.

Line 57: Please clarify what you mean by "non-adherance to treatment." What treatment? Obesity treatment? Depression treatment? Medical care more generally? Also, to reduce stigma, I recommend something like "difficulty adhering to treatment recommendations" in order to reduce blame towards the person.

Authors’ response: Thank you for your comment. Further details regarding the non-adherence have been added in this sentence (lines 59-60).

Line 58: The authors generally do a really nice job of using person-first language in this manuscript. Although "eutrophic" is not a stigmatizing term in the same way that "obese" can be, I would still use person-first language here, e.g., "Compared to individuals with a BMI in the normal range."

 Authors’ response: Thank you for pointing this out and for suggesting a more neutral terms. The sentence has been corrected accordingly, on line 62.

In the paragraph starting on Line 101 that describes prior cluster analyses, I would be cautious about making it sound like all individuals with overweight or obesity have significant psychopathology. I might add some nuance to this paragraph to clarify that this is not the case and to reiterate that the reason for examining these clusters is to ensure that individuals with obesity with various psychological needs are adequately supported.

Authors’ response: Thank you for your suggestion. Additional information were provided in this paragraph (lines 106-108).

Other comments:

In Figures 1 and 2, it would be helpful if the labels matched the way that you described the measures in the text (e.g., Calling "App" SSES-app or something like that). Otherwise, it's a bit hard to switch between the figure and the text.

Authors’ response:  Thank you for your advice. We have taken yours and the second reviewer’s suggestions into account and he figures have been modified accordingly.

Round 2

Reviewer 1 Report

I have no further comments.

Author Response

Dear Reviewer,
Thank you for your valuable comments. 

Reviewer 2 Report

The authors have implemented most of the notes well. Only the graphics seem unchanged in the manuscript, perhaps a modified figure version is uploaded separately? Otherwise, the axis labelling should really be enlarged, the graphic is hardly readable. 

Author Response

Dear Reviewer, thank you for your advice. We have integrated your comments and have modified the figures in order to facilitate their interpretation. Hence, thicker lines, enlarged axis labelling and variables’ full names were added, as well as a black horizontal line that represents the value 0, so that the lector can easily spot the mean.